# Next-Generation Sequencing-Based Monitoring of Intestinal Bacteria and Bacteriophages Following Fecal Microbiota Transplantation in Inflammatory Bowel Diseases

**DOI:** 10.3390/pathogens12121438

**Published:** 2023-12-11

**Authors:** Oleg V. Goloshchapov, Oksana B. Shchukina, Aleksey V. Kusakin, Viktoria V. Tsai, Roman S. Kalinin, Yury A. Eismont, Oleg S. Glotov, Alexei B. Chukhlovin

**Affiliations:** 1R.M. Gorbacheva Memorial Institute of Oncology, Hematology and Transplantation, Pavlov First Saint Petersburg State Medical University, 197022 St. Petersburg, Russia; burmao@gmail.com (O.B.S.); alexei.chukh@mail.ru (A.B.C.); 2Pediatric Research and Clinical Center for Infectious Diseases, 197022 St. Petersburg, Russia; kusakin@scamt-itmo.ru (A.V.K.); pancu43@gmail.com (R.S.K.); y-eis@inbox.ru (Y.A.E.); olglotov@mail.ru (O.S.G.); 3Serbalab Laboratory, 199106 St. Petersburg, Russia; viktoriya14054@gmail.com

**Keywords:** gut microbiome, next-generation sequencing, bacteria, phages, inflammatory bowel disease, graft-versus-host disease

## Abstract

Inflammatory bowel diseases (IBD) and acute graft-versus-host disease (GVHD) are associated with persistent intestinal dysfunction preceded by gut bacterial dysbiosis. There are limited data on intestinal bacteriophages in these conditions. The aim of the present work was to detect associations between dominant intestinal bacteria by means of 16S rRNA gene sequencing, and some clinically significant viruses detected with a customized primer panel for NGS-based study. The clinical group included patients with Crohn’s disease (IBD, *n* = 9), or GVHD (*n* = 6) subjected to fecal microbiota transplantation (FMT) from healthy donors. The stool specimens were taken initially, and 5 times post-FMT until day 120. Using NGS approach, we have found a higher abundance of *Proteobacterota* phylum in GVHD, especially, at later terms post-FMT. Moreover, we found an early increase of *Klebsiella* and *E. coli/Shigella* abundance in GVHD, along with decreased relative content of *Faecalibacterium*. Upon evaluation of intestinal phageome, the relative amount of *Caudoviricetes* class was higher in GVHD. A significant correlation was found between *Proteobacteria* and *Caudoviricetes*, thus suggesting their association during the post-FMT period. Moreover, the relative amounts of five *Caudoviricetes* phage species showed distinct correlations with *Klebsiella* and *Enterococcus* ratios at different terms of FMT. In conclusion, parallel use of 16S rRNA gene sequencing and targeted NGS viral panel is a feasible and useful option for tracing specific viral strains in fecal microbiota. The developed array of viral primers may be extended to detect other phages infecting the clinically relevant bacteria.

## 1. Introduction

Over recent years, a number of studies was performed on the fecal microbiome profile and its possible role in inflammatory bowel disorders (IBD) [1,2]. In particular, Crohn’s disease and ulcerative colitis are associated with persistent intestinal dysfunction and malnutrition [3]. Along with inflammatory lesions of intestinal mucosa, significant shifts in gut microbiota are observed in these disorders [4]. Acute graft-versus-host disease (GVHD) occurs after allogeneic stem cell transplantation (allo-HSCTs) and intensive anti-infectious treatment being also complicated by lesions of intestinal mucosa and microbial dysbiosis [5].

Fecal microbiota transplantation (FMT), along with rational antibiotic therapy, are considered effective strategies of intestinal microbiota correction [6]. However, broad variability of microbiome composition in the patients with gut disorders suggests a search for clinically relevant microorganisms which could be monitored and modulated by appropriate therapies.

The intestinal inflammation is commonly associated with some local commensal microorganisms of *Proteobacteria* phylum, e.g., *Klebsiella* spp., *Pseudomonas* spp., *E. coli/Shigella* etc. Relative abundance of these bacteria in the microbiome may be assessed by means of next-generation sequencing (NGS). Meanwhile, there are limited data on prokaryotic viruses (bacteriophages) infecting these microorganisms [2]. In genetic terms, the intestinal phageome still represents a ‘dark matter’ due to absence of reference sequences to classify them at the species level, like as 16S rRNA gene in bacteria. Among bacteriophages, *Caudovirales* seem to predominate in the intestinal microbiota [7]. Hence, the aim of the study was to detect associations between commensal intestinal bacteria assessed by 16S rRNA gene sequencing, and distinct bacteriophages using a customized primer panel for significant relevant viruses. We have found an association between relative contents of *Klebsiella*, *Enterococci*, and their specific phages following fecal microbiota transplantation (FMT) in the patients with gut inflammatory conditions.

## 2. Materials and Methods

### 2.1. Ethical Statement

All methods of collecting biological samples from stool used under this medical examination were taken with the approval of the attending physician. The study was conducted in accordance with guidelines of the 1964 Declaration of Helsinki and its later amendments. All patients or their guardians signed a written informed consent form for a hematopoietic stem cell transplant and the subsequent medical procedures, as well as potential usage of their clinical data for the purposes of clinical research. This study was approved by the Local Review Board of the Pavlov First State Medical University of St. Petersburg (ID number 214 of 17 December 2018).

### 2.2. Patients and Study Design

The study included 15 adult patients subjected to fecal microbiota transplantation (FMT). These patients presented a limited group treated at the Intensive Care Unit of I. Pavlov Medical University due to severe, life-threatening intestinal syndrome with diarrhea, and pain being resistant to conventional therapies. Nine patients were observed at the Department of General Medical Practice (Pavlov University), being diagnosed with inflammatory bowel disorder (IBD) which was clinically and morphologically confirmed as Crohn’s disease (Table 1). These patients suffered with severe colitis dependent on steroid therapy. Clinical indications for FMT included progression of the disease and its resistance to steroid therapy. Moreover, six patients were treated for drug-resistant graft-versus-host disease (GVHD) after allo-HSCT performed at the R.Gorbacheva Memorial Institute of Pediatric Oncology Hematology and Transplantation. Primary hematological disorders in this group were as follows: acute leukemias (*n* = 3), chronic myeloid leukemia (*n* = 2), and aplastic anemia (*n* = 1). Before allo-HSCT, the patients underwent anticancer therapy causing malnutrition and weight loss enhanced by intestinal GVHD posttransplant (Table 1). A group of four healthy subjects served as donors for these patients. The donors of fecal microbiota were evaluated and screened according to the European Guidelines on faecal microbiota transplantation [8]. In brief, the candidate donors should have normal blood counts and serum biochemistry, negative tests for HIV, hepatitis viruses (HVA to HVE), syphilis. Donor stool was examined by routine bacteriological methods for common fecal bacteria, especially, *Shigella* group; for the strains resistant to antibiotics. Microbial profile of the donor stool was also examined for the most common bacteria using a multiplex PCR panel (Colonoflor-16, LLC Alfa Lab, Saint Petersburg, Russia). The donor stool samples were tested for enteropathogenic viruses by PCR. The donor testing included protozoa and helminthes, studies of fecal calprotectin. To correct gut microbiota, all patients were subjected to fecal microbiota transplantation (FMT) performed from healthy donors according to standardized protocol [9]. In brief, the treatment consisted of administration of 15 gelatine capsules on 2 consecutive days (a total of 22 g/30 capsules) in absence of anti-infectious therapy within preceding 2 weeks.

The stool samples were collected prior to FMT (day 0) as well on days +3, +15, +30, +60. A total of 70 fecal specimens have been processed for further NGS of 16S rRNA gene.

Moreover, a primer panel was developed and applied for NGS-based detection of common eukaryotic viruses and bacteriophages infecting some clinically relevant intestinal microbiota, e.g., *Klebsiella* spp., *Pseudomonas* spp., *E. coli/Shigella*, *Salmonella*, etc. (see Appendix A [10,11,12,13,14,15,16,17,18]).

### 2.3. DNA Isolation and NGS Procedures

Total DNA was isolated from the suspended stool samples subjected to homogenization in a lysing solution homogenized with the bead technique, followed by DNA extraction by the sorbent column technique (Qiagen, Germantown, MD, USA) according to the manufacturer’s recommendations.

The 16S DNA sequencing libraries were prepared according to Illumina’s 16S Metagenomic Sequencing Library Preparation protocol (Part #15044223 Rev. B). The reagent kits were purchased from Illumina (Illumina, San Diego, CA, USA). We used 5 ng of total DNA per sample in order to amplify the target fragment of the 16S rRNA bacterial gene by means of the recommended primers for the V3–V4 region. To detect viruses, we applied our customized primer panel for a series of dsDNA primers described elsewhere (Appendix A [10,11,12,13,14,15,16,17,18]).

We performed 25 PCR cycles using the KAPA HiFi HotStart ReadyMix (2X) (Roche Diagnostics, Zug, Switzerland). After purification of PCR products with the SPRI bins, we indexed 5 ng of the resulting amplicons with the KAPA HiFi HotStart ReadyMix (2X) (Roche Diagnostics, Switzerland) and the Nextera XT Index Kit (Illumina, USA). We ran 8 cycles of index PCR according to the Illumina protocol. The obtained libraries were sequenced using the Illumina MiSeq platform.

### 2.4. Bioinformatic Analysis of Bacterial 16S rRNA

The microbial composition of the samples was analyzed using the VSEARCH software (version v2.22.1) [19]. Paired-end reads were merged using the ‘–fastq_mergepairs’ option. The parameters used included a minimum overlap length of 20 bases and a maximum of 10 differences. Quality statistics for the merged reads were calculated with the ‘–fastq_eestats’ option in VSEARCH. Sequences with error rates exceeding 1.0%, shorter than 400 bases, or containing unknown bases (N) were removed using the ‘–fastq_filter’ command. Dereplication was conducted with the ‘–derep_fulllength’ function to collapse identical sequences into unique representatives, setting a minimum unique size of 2. Chimeric sequences were identified and removed using reference-guided chimera detection with the RDP “Gold” database, and the parameters included ‘–uchime_ref’. Non-chimeric sequences were retained.

The operational taxonomic units (OTUs) were defined at a 97% sequence similarity threshold using the ‘–cluster_size’ function in VSEARCH. OTU centroids and a count table were generated. Sequences were clustered based on a 97% similarity threshold using ‘–id 0.97’. Taxonomy was assigned to the generated OTUs using the Ribosomal Database Project (RDP) V16 database [20] with VSEARCH. The cutoff for taxonomy assignment was set at 0.7, and the output included the OTU ID and the taxonomic lineage.

To compare the relative contents of bacterial phyla and genera in GVHD and IBD groups, we used standard STATISTICA 5.0 software. The descriptive results for distinct groups were expressed as M ± m. The inter-group differences were evaluated by parametric methods (*t*-test). Correlation quotients were assessed by means of a non-parametric Spearman criterion. The confidence levels of *p* < 0.05 were considered statistically significant. Statistical differences in abundance of bacterial taxa for different groups were assessed by means of non-parametric Mann–Whitney U test for paired comparisons.

### 2.5. Classification of Detected Viral Sequences

The search and classification of viral sequences was performed on the results obtained after applying a custom primer panel (Appendix A [10,11,12,13,14,15,16,17,18]). Prior to classification, a quality control of the reads was performed, using the FastQC tool (v0.11.9) [21]. Adapters and low quality reads were removed using the Trimmomatic software (v0.39) [22] with the following parameters ‘LEADING:30’, ‘TRAILING:30’, ‘SLIDINGWINDOW:10:28’, ‘MINLEN:36’.

The classification was achieved through the utilization of two key bioinformatics softwares, Kraken 2 and Kaiju [23,24], each serving to provide taxonomic identification and abundance estimation in metagenomic data.

In the Kraken 2 pipeline, the standard Kraken 2 database of nucleotide sequences served as the reference resource. In addition, the ‘—minimum-hit-groups 3’ parameter was used to set the minimum number of hit groups equal to 3 that must be found to declare the sequence classified. Moreover, the Bracken was applied to estimate the abundance of classified sequences to the species level. Visual representation of the results was realized through the generation of Krona HTML plots derived from the Bracken reports.

Unlike the pipeline for Kraken 2, the Kaiju pipeline employs the viral database of peptide sequences (NCBI RefSeq viral database [25]). The Kaiju tables, focusing on the species level, were generated for each sample, providing quantitative insight into the abundance of viral taxa. To enhance the clarity and interpretability of the outcomes, Krona reports were generated, offering visual representation in HTML format.

## 3. Results

### 3.1. Post-FMT Changes for Clinically Relevant Bacteria

Sufficient temporal dynamics is shown for phylum *Proteobacterota* (A). Its ratio in fecal microbiota of GVHD patients tended to increase compared with IBD patients from day 3 to day 30 after FMT (*p* = 0.23), as seen in Figure 1. Of note, *Proteobacterota* include a variety of clinically important bacteria causing severe infectious complications in immunocompromised patients (*Klebsiella*, *Pseudomonas*, *Acinetobacter*, *E. coli* and *Shigella*) thus suggesting FMT to be a microbiome- correcting treatment.

Other major phyla of stool microbiota (*Bacillota*, *Bacteroidota*, *Actinomycetota*) did not differ between IBD and GVHD at any time point pre- and post-FMT.

Among clinically relevant bacterial genera, *Klebsiella* spp. was the main object in our study. It was nearly absent (about 0.1% of total microbiota) in the stool of control subjects (Figure 2A). In patients before FMT, the relative content was increased against healthy controls (3 to 6%; *p* < 0.01), both in IBD and GVHD groups. In IBD cases, this ratio remained low during the entire post-FMT period. In GVHD patients, a sufficient increase of *Klebsiella* contents over control values was revealed on days +3 and +30 after FMT, thus suggesting its persistence over time.

Relative contents of *E. coli/Shigella* (Figure 2B), were increased against controls before FMT (3.8 ± 1.4% vs 0.06 ± 0.04; *p* < 0.02) as well as on days 3 to 30 (7 to 12%). In IBD, these bacterial species remained low up to 60 days after FMT (a mean of 1.5 to 3%).

We gave also assessed the time changes in two bacteria of *Bacillota* phylum. The ratio of *Faecalibacterium* in IBD patients did not differ from healthy controls (8.1 + 3.5%), whereas in GVHD patients it tended for lower levels over 60 days pre- and post-FMT (*p* = 0.01) as seen from Figure 2C. Meanwhile, *Enterococcus* ratio in IBD patients was close to minimal control levels (about 0.1% of total bacterial mass), being, however increased in some GVHD cases before FMT (a mean of 10%) then being decreased to healthy controls. Hence, certain bacterial genera exhibit individual changes after FMT, depending on the origin of underlying intestinal disorder.

### 3.2. Phageome Changes Following FMT

Using a customized primer viral panel at the NGS platform, we have detected a number of prokaryotic viruses and traced their temporal changes after FMT. Among them, *Caudoviricetes* is the most studied in clinical settings.

Firstly, we compared the detection efficiency of intestinal viruses by two standard taxonomic classification systems, i.e., nucleotide-based Kraken 2 system, and peptide-based Kaiju database. Both systems showed a sufficient ratio of non-classified sequences (>15%) at the phylum and genus levels. However, the total number of identifiable viral species was much higher with Kaiju classifier than when using Kraken 2 system (41 versus 10 species, respectively).

The difference in *Caudoviricetes* ratio between IBD and GVHD proved to be better discerned with Kaiju than with Kraken 2 database showing higher abundance of this class of phages in GVHD than in IBD patients over post-FMT period (Figure 3A,B). Relative contents of *Caudoviricetes* did not differ between IBD and GVHD pre-TFM, however, being permanently lower in IBD patients (*n* = 9) versus GVHD (*n* = 6) on D+1 to D+30 after fecal transplantation (*p* = 0.03).

*Lederbergvirus*, a genus of *Caudovirales*, also exhibited higher discrimination between GVHD and IBD cases at initial time point (Figure 3C,D), both attaining minimal levels at later terms (up to 60 days post-FMT).

Within first days after FMT, a sufficient decrease was revealed for some bacteriophages which infect the clinically actual intestinal bacteria. E.g., we observed a decreased ratio of *Enterobacter* phage IME, *E. coli* phage mEpX2, and *E. coli* phage HK106 in total group of patients compared to the pre-FMT time point. These intestinal viruses belong to *Caudoviricetes* phages and are hosted by the Gram-negative bacteria (Table 2). The members of *Crassvirales* were not considered in our further analysis since they encode still uncultured phages [26].

Mastadenovirus D (type 54) was found in a single patient on the first days after FMT, being, however, not detectable at later terms. In other patients, our viral NGS array did not reveal any DNA viruses including herpesviruses which were also included into this panel. A study of these DNA samples by clinical multiplex PCR system (Intestinal panel, Amplisens, Moscow, Russia) did not also reveal any DNA viruses, including adenovirus (this test system detects serotype 40/41 of AdV).

### 3.3. Correlations between Bacterial Groups and Detectable Bacteriophages

We have not found any positive correlations between the percentage of detectable phages and major bacterial phyla, except of significant correlation between *Proteobacteriota* phylum and bacteriophages of *Caudoviricetes* class (Table 3). However, using Kaiju database for alignment of NGS data, we were able to reveal positive correlations between some clinically important *Enterobacteria* and ratios of several bacteriophages found in these samples.

## 4. Discussion

Fecal microbiota transplantation have shown its efficiency in restoring both bacterial and viral microbiota in patients with *C. difficile*-associated gut dysbiosis [28]. The authors used a common NGS metagenomic study showing significant changes of major bacterial and viral classes post-FMT. Similar interplay between bacteriome and virome may be revealed in gastrointestinal disorders following allogeneic transplants of hematopoietic stem cells [29].

Experience with FMT in Crohn’s disease was summarized in recent meta-analysis [30]. Clinical remission was revealed in about a half of patients treated by FMT, with minimal side effects related to the procedure. Assessment of intestinal microbiome showed an increased biodiversity and a general shift towards donor-like pattern of microbiota. Several clinical studies on FMT efficiency in steroid-resistant GVHD were recently discussed demonstrating complete clinical response in 40% of cases, with rare infectious complications [31]. When summarizing results from small clinical studies, the post-FMT samples show partial restoration of gut microbiota composition, in particular, *Ruminococcaceae*, *Lachnospiraceae*, *Bacteroidaceae*, *Streptococcaceae*, and *Lactobacillaceae* [32]. In our experience, clinical response in refractory GVHD is associated with increased contents of *Bacteroides fragilis* and *Faecalibacterium prausnitzii* in fecal samples at 2–4 weeks after FMT [33].

In this respect, metagenomic assays cover broad variety of infectious pathogens. Meanwhile, usage of targeted NGS primer panels for microbiome studies enabled us to detect within *Caudoviricetes* some distinct bacteriophage strains in faeces prior to FMT. Presence and contents of these individual bacteriophage strains showed significant positive correlations with some clinical relevant facultative anaerobes of *Proteobacterota* phylum, especially, *Klebsiella*, *E. coli* and *Salmonella*. These findings provide a tool for searching and tracing certain *Enterobacteria* phage strains. Along with fecal microbiota transplantation, the phage therapy may potentially prevent colonization with antibiotic-resistant bacteria [34].

Upon extension of customized bacteriophage panel, the proposed NGS-based approach will provide detection and monitoring of other phage strains, especially, those infecting *Enterobacteriales*. Similar approach may be used for the group of phages associated with *Enterococcus* (*Lactobacteria*). Moreover, some correlations were found between certain *Firmicutes/Bacillota* (*Faecalibacter*, *Merdibacter*, *Roseburia*) and great classes of bacteriophages (*Preplasmaviricota*, *Peploviricota*) thus confirming existence of specialized phage communities which infect these strictly anaerobic bacteria. So far, the *Firmicutes*-specific phages are poorly studied due to problems with phage biotesting in anaerobic bacterial cultures [4]. Metagenomic studies of intestinal virome allow to trace some common interactions between distinct members of bacteriome and virome in Crohn’s disease [35]. The data from such metagenomic approach suggested a participation of bacteriophages of *Roseburia* in pathogenesis of metabolic syndrome [36]. We have limited a search for phages by a panel for detection of the most important viral sequences, finding a number of phage strains which are likely to infect clinically relevant gut bacteria in the individual patients. Interestingly, most of the *Klebsiella*-associated phages are detected at pre-transplant terms, then disappear. The reasons for these changes may be as follows: enhanced bacteriolytic effects of phages after transplantation of healthy fecal material; exhaustion (decolonization) of the phage-bearing bacterial strains after FMT.

Metagenomic studies of DNA from environmental specimens allow to detect multiple relations between bacteria and viruses, e.g., possible horizontal transfer of antibiotic resistance genes [37]. In this respect, the NGS approaches, both shotgun metagenomics, and usage of customized primer panels should promote identification of bacteriophage strains specific for anaerobic gut microorganisms.

Our pilot study included parallel profiling of intestinal bacteria and selected viruses using NGS-adapted primer panels. The patients in our study were, mostly, adults. Despite broad scatter of microbiota composition, its main phyla and biodiversity in adult group are at comparable ratios [38].A limitation of our work is small number of patients with GVHD and Crohn’s disease available for FMT in our setting. Moreover, we used a restricted set of virus-specific primers for initial version of customized NGS panel. Extension of the primer panel, especially for the phages of strictly anaerobic bacteria, will be added to the NGS primer set in future studies on bacterial and viral landscape before and after FMT.

## 5. Conclusions

The developed primer panel for NGS of intestinal DNA viruses enabled us to detect and trace temporal dynamics of distinct bacteriophages in faecal samples from the patients with IBD and GVHD treated by fecal microbiota transplantation.

Intestinal dysbiosis in IBD and graft-versus-host disease (GVHD) is characterized by expansion of *Proteobacteria* in fecal material in cases of GVHD, along with changes in several phage populations.

Parallel usage of 16S rRNA gene sequencing and targeted viral panel for NGS may be a suitable option for tracing different classes of specific viral strains in fecal microbiota. The developed array of gene-specific viral primers may be extended to detect different bacteriophages infecting the clinically relevant bacteria.

## Figures and Tables

**Figure 1 pathogens-12-01438-f001:**
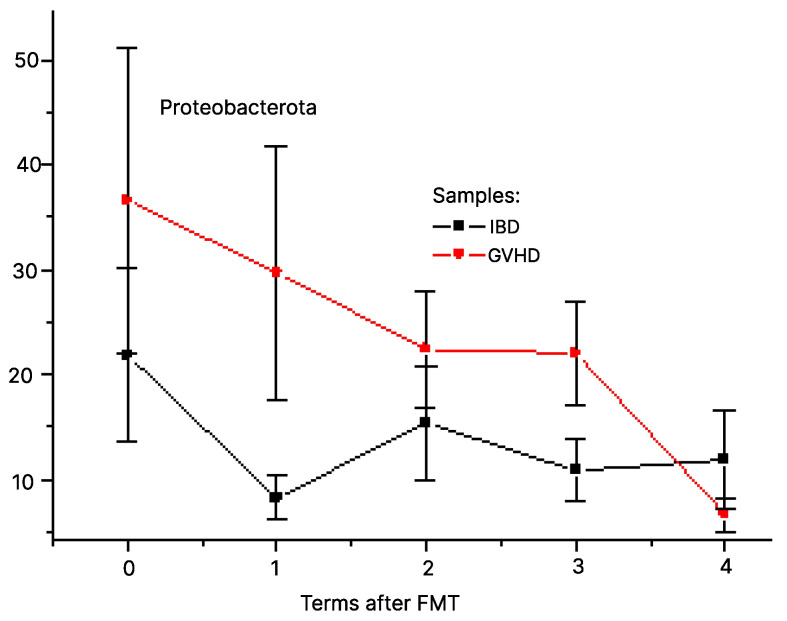
Relative contents of Proteobacterota after FMT in patients with IBD (black squares) and GVHD (red circles). Abscissa, terms after FMT (0 to 4), respectively, pre-FMT (0); days +3 (1); +15 (2); +30 (3); +60 (4)). Ordinate, per cent of the given taxon in total bacterial mass.

**Figure 2 pathogens-12-01438-f002:**
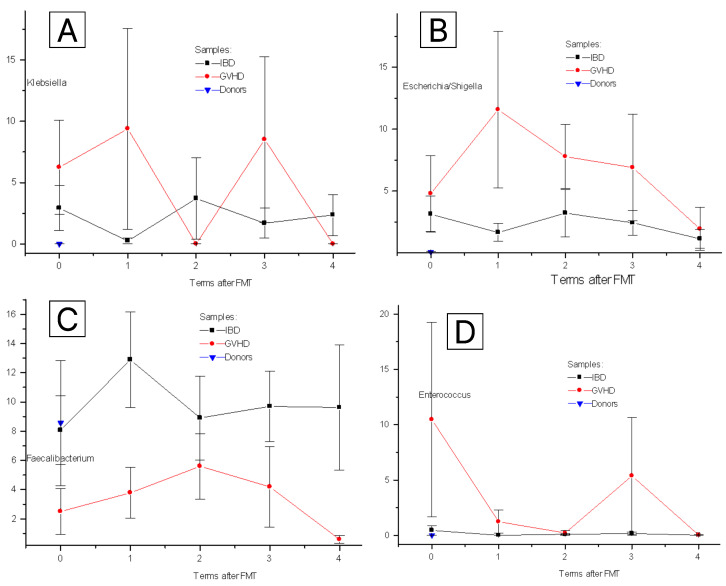
Time-dependent changes of relative bacterial contents after FMT in patients with IBD (black squares), GVHD (red circles) and healthy controls (blue triangles). Sufficient temporal dynamics is shown for some clinically relevant bacterial genera: *Klebsiella* (**A**), *E. coli/Shigella* (**B**), *Faecalibacterium* (**C**), *Enterococcus* (**D**). Abscissa, terms after FMT (0 to 4), respectively, pre-FMT (0); days +3 (1); +15 (2); +30 (3); +60 (4)). Ordinate, per cent of the given bacterial taxon in total bacterial mass.

**Figure 3 pathogens-12-01438-f003:**
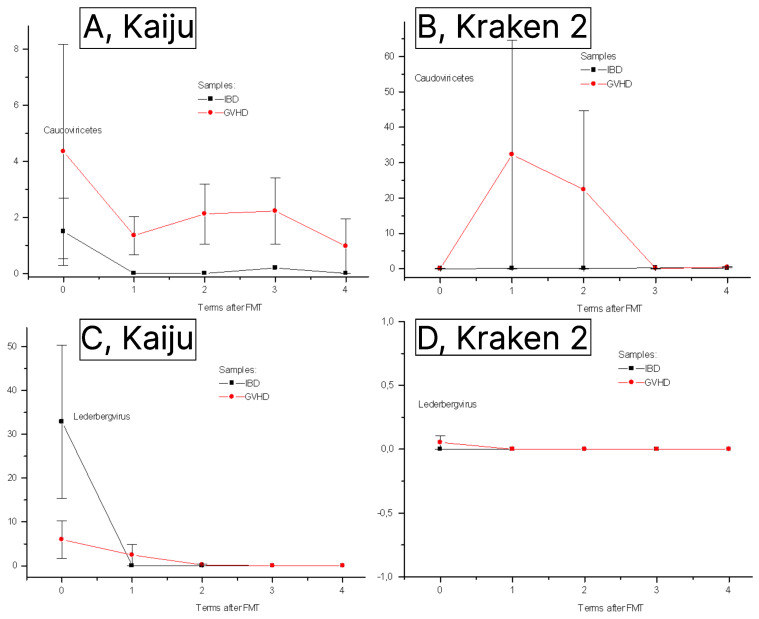
Comparative detection levels of bacteriophage groups using Kracken and Kaiju classification systems ((**A**,**B**)—*Caudoviricetes*; (**C**,**D**)—*Lederbergvirus*). Abscissa, terms after FMT (0 to 4), respectively, pre-FMT (0); days +3 (1); +15 (2); +30 (3); +60 (4). Ordinate, per cent of major phage groups in a variety of DNA sequences.

**Table 1 pathogens-12-01438-t001:** Demographic and clinical characteristics of the patients subjected to FMT.

Parameters	Crohn’s Disease (*n* = 9)	Graft-Versus-Host Disease (*n* = 6)
Gender (males/females)	5/4	4/2
Median age, years (min and max values)	29 (22–59)	27 (9–59)
Body mass, kg (min and max values)	66 (42–69)	49 (28–70)
Duration of the primary illness (min and max values)	4 (1–15)	6 (2–9)

**Table 2 pathogens-12-01438-t002:** A spectrum of intestinal viruses detected in the study group classified by the NCBI Browser [27].

Phylum	Class	Genus	Species	Bacteriophage
Uroviricota	Caudoviricetes	Bievrevirus	Bievrevirus bv4A7	Escherichia phage 4A7
		Tequatrovirus	Tequatrovirus gee4498	Escher phage vB_EcoM_G4498
		Traversvirus	Traversvirus II	Escherichia phage Stx2 II
		Shamshuipovirus	Shamshuipovirus mEpX2	Escherichia phage mEpX2
		Eganvirus	NC	Salmonella phage BIS20
		Brunovirus	Brunovirus SEN34	Salmonella phage SEN34
		Goslarvirus	Goslarvirus goslar	Escherichia phage vB_EcoM_Goslar
		Lambdavirus	NC	Enterobacteria phage mEp237
		Lederbergvirus	Lederbergvirus Sf6	Shigella phage Sf6
		Lederbergvirus	NC	Enterobacter phage IME10
		Lederbergvirus	NC	Salmonella phage SEN22
		Novemvirus	Novemvirus T5282H	Enterobacter phage phiT5282H
		Oengusvirus	Oengusvirus oengus	Faecalibacterium phage FP_oengus
		Caudoviricetes unclassified	NC	Aeromonas phage pAEv1818
Uroviricota	Caudoviricetes (Crassvirales)	Cohcovirus	Cohcovirus splanchnicus	uncultured phage cr30_1
		Kahnovirus	Kahnovirus copri	uncultured phage cr44_1
		Kahnovirus	Kahnovirus oralis	uncultured phage cr85_1
		Birpovirus	Birpovirus hominis	uncultured phage cr19_1
		Jahgtovirus	Jahgtovirus intestinihominis	uncultured phage cr107_1
Peploviricota	Herviviricetes	Cytomegalovirus	Human betaherpesvirus 5	NC
Preplasmiviricota	Tectiliviricetes	Mastadenovirus	Mastadenovirus D	Mastadenovirus 54 (serotype)

**Note:** NC, non-classifiable by NCBI Browser.

**Table 3 pathogens-12-01438-t003:** Corrrelations between the relative contents of bacteria (NGS of 16s rRNA gene) and certain bacteriophages in fecal DNA from IBD and GVHD patients (total group, 15 cases, 24 points).

Bacterial Phyla, Genera	Phages (Phyla, Genera, Species)	Correlation Quotients	*p* Value
Proteobacteriota	Caudoviricetes	0.667	0.0002
	Lederbergvirus	0.477	0.009
	Enterobacter phage IME10	0.552	0.004
	*E. coli* phage 4A7	0.617	0.0065
	Salmonella phage SEN34	0.538	0.003
	Aeromonas phage pAEv1818	0.535	0.004
Genus Faecalibacteria	Bamfordviricota, Preplasmaviricota	0.730	0.00003
Clostridia_ss	Peploviricota	0.660	0.0002
Klebsiella	Lederbergvirus	0.703	0.00006
	Enterobacter phage IME10	0.625	0.0055
	*E. coli* phage HK106	0.567	0.002
	Traversvirus	0.526	0.004
	Salmonella phage SEN34	0.564	0.002
	Aeromonas phage pAEv1818	0.561	0.002
Enterococcus	Caudoviricetes	0.630	0.0005
	*E. coli* phage 4A7	0.479	0.009
	Salmonella phage SEN34	0.489	0.007
	Aeromonas phage pAEv1818	0.504	0.006

**Notes:** Non-parametric Spearman criterion was used to assess correlation quotients and confidence levels.

## Data Availability

We report no links to publicly archived datasets analyzed or generated during the study.

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
