# Peer review of "Next-Generation Sequencing-Based Monitoring of Intestinal Bacteria and Bacteriophages Following Fecal Microbiota Transplantation in Inflammatory Bowel Diseases"

_pathogens, 2023, doi:10.3390/pathogens12121438_

Round 1

Reviewer 1 Report

Comments and Suggestions for Authors

The manuscript presents an original study highlighting the impressive changes of major bacterial and viral classes after FMT and their association with clinical improvement post-FMT. Therefore, the study focuses on the interplay between bacteriome and virome after FMT and the topic is of great interest to both clinicians and researchers focusing on fecal microbiota transplantation. The current manuscript has a significant level of novelty and has a particular strength in tracing of the temporal dynamics of distinct bacteriophages in faecal samples from the patients who received FMT. Although the parallel inclusion and follow-up of CD and aGVHD patients after FMT is unusual, considering several aspects, such as the different mechanisms leading to dysbiosis and the immunological context, I find the choice of methodology interesting.

The authors should include a description of the donor screening process for FMT donors in the methods section and also include a descriptive statistics section with patients’ characteristics in the beginning of the results section. A limitation of the study is the very small number of included patients, which limits the generalization of results, especially considering that some patients were paediatric patients, therefore increasing heterogeneity. These aspects should be acknowledged in the discussion section, while also mentioning other limitations of the presented research.

Minor comments:

Please remove the words “non-specific” before mentioning the diagnosis of ulcerative colitis in the introduction section.

Comments on the Quality of English Language

Minor editing of English language required

Author Response

Dear Reviewer,
Much thanks for careful work. We tried to answer Your queries as follows:

Materials and Methods
Clinical characteristics of patients are presented in detailed form (lines 60-73), and Table 1 is added to Materials and Methods.

Donor screening process
Line 73: The donors of fecal microbiota were evaluated and screened according to the European Guidelines on faecal microbiota transplantation [8]. In brief, the candidate donors should have normal blood counts and serum biochemistry, negative tests for HIV, hepatitis viruses (HVA to HVE), syphilis. Donor stool was examined by routine bacteriological methods for common fecal bacteria, especially, Shigella group; for the strains resistant to antibiotics. Microbial profile of the donor stool was also examined  for the most common bacteria using a multiplex PCR panel (Colonoflor-16, LLC Alfa Lab, Russia). The donor stool samples were tested for enteropathogenic viruses by PCR. The donor testing included protozoa and helminthes, studies of fecal calprotectin.

One reference is added:
8. Cammarota, G.; Ianiro, G.; Tilg, H.; Rajilić-Stojanović, M,; Kump, P.; Satokari, R.; Sokol, H.; Arkkila, P.; Pintus, C.; Hart, A.; et al. European consensus conference on faecal microbiota transplantation in clinical practice. Gut 2017,66,569. doi: 10.1136/gutjnl-2016-313017.

Descriptive statistics:
A section on descriptive statistics is introduced (Line 128)

Discussion: limitations of the study
A limitation of our work is small number of patients with GVHD and Crohn’s disease available for FMT in our setting. Moreover, we used a restricted set of virus-specific primers for initial version of customized NGS panel. Extension of the primer panel, especially for the phages of strictly anaerobic bacteria, will be added to the NGS primer set in future studies on bacterial and viral landscape before and after FMT. (line 274)

Minor comments:
Please remove the words “non-specific” before mentioning the diagnosis of ulcerative colitis in the introduction section.
– ‘non-specific’ is removed

Reviewer 2 Report

Comments and Suggestions for Authors

Dear Colleagues,

I have completed a thorough review of the manuscript titled "NGS-based monitoring of intestinal bacteria and bacteriophages following fecal microbiota transplantation in the patients with inflammatory bowel disorders" submitted to Pathogens. The authors have conducted significant research, employing 16S rRNA gene sequencing to detect associations between dominant intestinal bacteria and clinically relevant viruses using a customized primer panel for NGS-based study. While the manuscript presents a strong foundation, there are specific areas that require attention and revision. My detailed comments are provided below:

-The introduction effectively establishes the context and importance of the research topic.

-The methods section is well-written and sufficiently detailed. However, the authors have included 9 IBD patients and 6 aGVHD patients in their study without providing essential details such as age, body mass index, and sex etc... Including this information, if available, would enhance the completeness of the study.

-The authors should clarify whether the 16S rRNA sequencing data will be deposited and made available to the public.

-Asterisks should be added to figures to denote significant differences, and labels should be included for the ordinate.

Minor typos should be addressed. For instance:

Line 26 and 30: "Intestinal Mucosae" should be changed to "Intestinal mucosa."

Line 99: Add a space between “differences.” and “quality.”

These revisions will strengthen the manuscript and enhance its scientific rigor.

Author Response

Dear Reviewer,
Much thanks for Your remarks that are answered as follows:

Line 65: …(Table 1). These patients suffered with severe colitis dependent on steroid therapy. Clinical indications for FMT included progression of the disease and its resistance to steroid therapy.

Line 69: Primary hematological disorders in this group were as follows: acute leukemias (n=3), chronic myeloid leukemia (n=2), and aplastic anemia (n=1). Before allo-HSCT, the patients underwent anticancer therapy causing malnutrition and weight loss enhanced by intestinal GVHD posttransplant (Table 1).

– The authors should clarify whether the 16S rRNA sequencing data will be deposited and made available to the public.

The 16S rRNA sequencing data do not contain any novel bacterial taxons, thus is not planned for deposition in public databases.

– Asterisks should be added to figures to denote significant differences, and labels should be included for the ordinate.

In cases of statistical significance, appropriate p values are indicated in the text. The labels for ordinate are indicated in the legends to figures (per cent of the given taxon in total bacterial mass).

Minor typos should be addressed. For instance:

Line 25 and 28: "Intestinal Mucosae" should be changed to "Intestinal mucosa"
Corrected

Line 114: Add a space between “differences.” and “quality.”
Corrected

Reviewer 3 Report

Comments and Suggestions for Authors

Congratulations to the authors for the study. The authors should be commended. It is relevant as the incidence of inflammatory bowel disease is increasing. The interest in the relationship between dysbiosis of gut microbiota and IBD has increased. Included below is a summary of areas that need to be reviewed and improved: 

1.       Title

a)      Replace the abbreviation in the title.

b)      Remove “the” before “the patients”.

c)       May drop “in the patients with” to make it ‘in inflammatory bowel disease’ to shorten the title.

d)      It is not every bowel disorders but Crohn’s disease and aGVHD.

2.       Abstract

a)      Common IBD is vague.

b)      Dysbiosis precedes intestinal dysfunction.

c)       Leave out “only” in most of the statements, as it is not adding value.

d)      Rephrase “clinically relevant”. Viruses.

e)      Write D+ 120 as Day 120.

f)        Choose one among aGVHD, GVHD and GvHD and use it consistently.

g)      Why E.coli/Shigella and not separate.

h)      Ratio should have at least 2 players.

i)         Revise or remove “sufficiently” in “sufficiently higher”.

j)         Conclusion: The study did not investigate suitability but usefulness.

3.       Keywords

a)      Write NGS in full.

b)      Cut down the overall number of words.

4.       Introduction

a)      Choose one and use consistently: non-specific IBD versus common IBD.

b)      Cite at least one reference after “The common inflammatory bowel diseases (Crohn’s disease and non-specific ulcerative colitis” are associated with persistent intestinal dysfunction and malnutrition.

c)       Key information in scientific writing should preferably not be placed within brackets i.e. remove the brackets around “Crohn’s disease and non-specific ulcerative colitis”.

d)      It should be ulcerative colitis or non-specific colitis and not “non-specific ulcerative colitis”.

e)      Specify at least one of the so-called “chronic conditions”.

f)        Use simply and direct language e.g. Acute graft versus host disease (aGvHD) instead of “ Another immune mediated intestinal disorder….”.

g)      Replace “exhaustion of dominant intestinal microbiota” with just dysbiosis.

h)      Same comment regarding E.coli/Shigela.

i)         Limit the use of words which do not improve understanding in the context they are used such as:  commonly, clinically, massive anti-infectious treatment, severe, total microbiome (microbiome), still only limited (is limited), seem to dominate, aim of the present work (aim of the study), hosted by these microorganisms,  clinically sound intestinal bacteria (commensal or non-pathogenic), clinically relevant  viruses (pathogenic), distinct association (association).

5.       Materials and Methods

a)      Revise and replace “Taken only”.

b)      The abbreviation FMT should have been introduced after the first mention of faecal microbial transplantion in the introduction section.

c)        Patients and Study design

-          Add an explanation of the sample size was determined.

-          Correct “Morever”.

d)      DNA isolation and NGS procedures: No concerns identified.

e)      Include statistical tests and package(s) used.

6.       Results

a)      Leave out etc. after Shigella.

b)      Figure 1

-          Consider adding actual values to the line graphs in Figure 1. And, do the same for all the line graphs.

c)       Revise “Klebsiella spp. was the central object”.

d)      Figure 2

-          The composition of donor microbiota is not part of comparison of temporal trend. The composition can be covered adequately in the text.

-          The paragraph immediately after Figure 2 is confusing. “Increase, remained low and increase regarding E.coli/Shigella.

-          Separate E.coli and Shigella in the writing.

-          Is it ratio or proportion/percentage.

-          Correlation as opposed to association is used for testing relationship between continuous data. Please check if doing correlation test wasappropriate and include it in the data analysis subsection of methodology.

7.       Discussion.

a)      Replace impressive in “study showing impressive changes”.

b)      Clinical effect was not was not covered in the methodology and results, should therefore not appear for the first time in the discussion section.

c)       Correct “page” strains in paragraph 3.

d)      Comparison with findings from previous studies should be included. What did other investigators found regarding changes of the bacteriome  FMT?

e)      Add limitations of the study to include sample size and possible variation of microbiota due to age differences.

8.       Conclusion

a)      Replace sufficient in “sufficient expansion”.

b)      Replace distinct in “distinct changes”.

9.       References

a)      No concern.

10.   Supplementary files: No obvious concern.

Comments on the Quality of English Language

The quality of English language needs to be improved. Consider leaving out vague words, etc. and e.g. in scientific writing. 

Author Response

Dear Reviewer,
We are much appreciated for attentive reading and hard editorial work. We tried to answer Your queries and added some required materials.

1.       Title
a)      Replace the abbreviation in the title.
b)      Remove “the” before “the patients”.
c)       May drop “in the patients with” to make it ‘in inflammatory bowel disease’ to shorten the title.
d)      It is not every bowel disorders but Crohn’s disease and aGVHD.

– The title is modified with respect to Your remarks

2.       Abstract
a)      Common IBD is vague.
– Line 1: ‘Common’ is removed

b)      Dysbiosis precedes intestinal dysfunction.
– Line 2: ‘and’ replaced by ‘preceded by’

c)       Leave out “only” in most of the statements, as it is not adding value.
– ‘only’ was skipped in most cases (lines 2, 40, 53, 128, 230)

d)      Rephrase “clinically relevant”. Viruses.
– Replaced by ‘significant’ viruses

e)      Write D+ 120 as Day 120.
– Replaced by ‘day 120’

f)        Choose one among aGVHD, GVHD and GvHD and use it consistently.
– GVHD is used throughout the text

g)      Why E.coli/Shigella and not separate.
– E.coli and Shigella species are closely related genetically, and are presented in 16S rRNA NGS reports in this way.

h)      Ratio should have at least 2 players.
– Line 10: ‘ratio’ replaced by ‘relative content’
– Line 11: ‘ratio’ removed

i) Revise or remove “sufficiently” in “sufficiently higher”.
– Line 12: ‘sufficiently’ removed

j)         Conclusion: The study did not investigate suitability but usefulness.
– Line 16: ‘suitable option’ is replaced by ‘feasible and useful option’

3.       Keywords
a)      Write NGS in full.
– Line 19: ‘next-generation sequencing’ instead ‘NGS’

b)      Cut down the overall number of words
– Line 19: ‘dysbiosis’ removed

4.       Introduction
a)      Choose one and use consistently: non-specific IBD versus common IBD.
– Lines 1, 24: ‘Common’ replaced by ‘non-specific’

b)      Cite at least one reference after “The common inflammatory bowel diseases (Crohn’s disease and non-specific ulcerative colitis” are associated with persistent intestinal dysfunction and malnutrition.
– The common inflammatory bowel diseases : replaced by ‘In particular’
– A citation is added to the reference list: [3]: Chang, J.T. Pathophysiology of inflammatory bowel diseases. New England Journal of Medicine 2020, 383, 2652-2664. https//doi.org/10.1056/NEJMra2002697.

c)       Key information in scientific writing should preferably not be placed within brackets i.e. remove the brackets around “Crohn’s disease and non-specific ulcerative colitis”.
– The brackets are removed

d)      It should be ulcerative colitis or non-specific colitis and not “non-specific ulcerative colitis”.
– Line 24: “non-specific “ removed

e)      Specify at least one of the so-called “chronic conditions”.
– Line 26: replaced by “… these disorders.”

f)        Use simply and direct language e.g. Acute graft versus host disease (aGvHD) instead of “ Another immune mediated intestinal disorder….”.
– Line 27-28: “Another … disorder….” skipped.

g)      Replace “exhaustion of dominant intestinal microbiota” with just dysbiosis.
– Line 28-29: “… exhaustion of …microbiota” replaced by “microbial dysbiosis”.

h)      Same comment regarding E.coli/Shigella.
– E.coli and Shigella species are closely related genetically, thus jointly presented in 16S rRNA NGS reports.

i)         Limit the use of words which do not improve understanding in the context they are used such as: commonly, clinically, massive anti-infectious treatment, severe, total microbiome (microbiome), still only limited (is limited), seem to dominate, aim of the present work (aim of the study), hosted by these microorganisms, clinically sound intestinal bacteria (commensal or non-pathogenic), clinically relevant viruses (pathogenic), distinct association (association).
– Line 29: “massive” replaced by “intensive”
– Line 30: “severe” removed
– Line 37: “total” replaced by “the”
– Line 40: “still” removed
– Line 39: “hosted by” replaced by “infecting”
– Line 43: “present work” replaced by “study”
– Line 43: “clinically sound” replaced by “commensal”
– Line 45: “relevant” changed to “significant”
– Line 45: “a distinct association” replaced by “associations”

5.       Materials and Methods

a)      Revise and replace “Taken only”.
– Line 53: “Only” removed

b)      The abbreviation FMT should have been introduced after the first mention of faecal microbial transplantion in the introduction section.
– Line 32: The (FMT) abbreviation is inserted

c)        Patients and Study design. Add an explanation of the sample size was determined.
– Line 65, Insertion following (FMT): …These patients presented a limited group treated at the Intensive Care Unit of I.Pavlov Medical University due to severe, life-threatening intestinal syndrome with diarrhea, and pain being resistant to conventional therapies.

          Correct “Morever”.
– Changed to ‘Moreover’

d)      DNA isolation and NGS procedures: No concerns identified.

e)      Include statistical tests and package(s) used.
– Line 128: To compare the relative contents of bacterial phyla and genera in GVHD and IBD groups, we used standard STATISTICA 5.0 software. The descriptive results for distinct groups were expressed as M + m. The intergroup differences were evaluated by parametric methods (t-test). Correlation quotients were assessed by means of a non-parametric Spearman criterion. The confidence levels of p < 0.05 were considered statistically significant. Statistical differences in abundance of bacterial taxa for different groups were assessed by means of non-parametric Mann–Whitney U test for paired comparisons.

6.       Results

a)      Leave out etc. after Shigella.
– ‘etc’ removed

b)      Figure 1
          Consider adding actual values to the line graphs in Figure 1. And, do the same for all the line graphs.
– Fig.1. is modified. If strongly required, we may present actual values on Fig.1 to 3 in special tables.
– Comments to Fig. 1 (lines 135-138) are also changed to: Its ratio in fecal microbiota of GVHD patients tended to increase compared with IBD patients from day 3 to day 30 after FMT (p=0.23), as seen in Fig.1.

c)       Revise “Klebsiella spp. was the central object”.
– Line 165: “Central” replaced by “main”

d)      Figure 2
          The composition of donor microbiota is not part of comparison of temporal trend. The composition can be covered adequately in the text.
– Concerning percentages of Klebsiella (line 166) and E.coli/Shigella (lines 172), we draw attention to increased pre-FMT values in GVHD and IBD against near-zero controls as seen from Fig.1A and 1B and described in text. Temporal trends should be defined in greater study groups.

          The paragraph immediately after Figure 2 is confusing. “Increase, remained low and increase regarding E.coli/Shigella.
– Lines 172: changed to: Relative contents of E.coli/Shigella (Figure 2B), were increased against controls before FMT (3.8 + 1.4% vs 0.06 + 0.04; p < 0.02) as well as on days 3 to 30 (7 to 12%). In IBD, these bacterial species remained low up to 60 days after FMT (a mean of 1.5 to 3%).

          Separate E.coli and Shigella in the writing.
– We define the E.coli/Shigella taxon as presented in 16S rRNA NGS reports (these species are closely related genetically).

          Is it ratio or proportion/percentage.
– ‘ratio’ changed to ‘percentage’
– The ordinate values represent percentages of distinct bacterial taxons in total bacterial mass (among total sequences detected by NGS in the given DNA sample).

          Correlation as opposed to association is used for testing relationship between continuous data. Please check if doing correlation test was appropriate and include it in the data analysis subsection of methodology.
– The correlations have been assessed for continuous data from paired series of NGS results. Correlation quotients were calculated using Spearman criterion (see Materials and Methods, line 131)

7.       Discussion

a)      Replace impressive in “study showing impressive changes”.
– Line 224: ‘impressive’ changed to ‘significant’

b)      Clinical effect was not was not covered in the methodology and results, should therefore not appear for the first time in the discussion section.
– Line 225: ‘associated with clinical improvement’ - removed

c)       Correct “page” strains in paragraph 3.
– ‘page’ changed to ‘phage’

d)      Comparison with findings from previous studies should be included. What did other investigators found regarding changes of the bacteriome FMT?
– Line 228: Experience with FMT in Crohn’s disease was summarized in recent meta-analysis [21]. Clinical remission was revealed in about a half of patients treated by FMT, with minimal side effects related to the procedure. Assessment of intestinal microbiome showed an increased biodiversity and a general shift towards donor-like pattern of microbiota. Several clinical studies on FMT efficiency in steroid-resistant GVHD were recently discussed demonstrating complete clinical response in 40% of cases, with rare infectious complications [22]. When summarizing results from small clinical studies, the post-FMT samples show partial restoration of gut microbiota composition, in particular, Ruminococcaceae, Lachnospiraceae, Bacteroidaceae, Streptococcaceae, and Lactobacillaceae [23]. In our experience, clinical response in refractory GVHD is associated with increased contents of Bacteroides fragilis and Faecalibacterium prausnitzii in fecal samples at 2-4 weeks after FMT [24].

e)      Add limitations of the study to include sample size and possible variation of microbiota due to age differences.
– Line 271: Our pilot study included parallel profiling of intestinal bacteria and selected viruses using NGS-adapted primer panels. The patients in our study were, mostly, adults. Despite broad scatter of microbiota composition, its main phyla and biodiversity in adult group are at comparable ratios [29].A limitation of our work is small number of patients with GVHD and Crohn’s disease available for FMT in our setting. Moreover, we used a restricted set of virus-specific primers for initial version of customized NGS panel. Extension of the primer panel, especially for the phages of strictly anaerobic bacteria, will be added to the NGS primer set in future studies on bacterial and viral landscape before and after FMT.

8.       Conclusion
a)      Replace sufficient in “sufficient expansion”
– ‘sufficient’ removed.

b)      Replace distinct in “distinct changes”.
– ‘distinct’ removed

9.       References
a)      No concern.

10.   Supplementary files: No obvious concern.

Round 2

Reviewer 3 Report

Comments and Suggestions for Authors

Thank you very much for the revised version. It is now understandable and easy to read. Well done to the authors.